# Improvement of the Polyhydroxyalkanoates Recovery from Mixed Microbial Cultures Using Sodium Hypochlorite Pre-Treatment Coupled with Solvent Extraction

**DOI:** 10.3390/polym14193938

**Published:** 2022-09-21

**Authors:** Gabriela Montiel-Jarillo, Diego A. Morales-Urrea, Edgardo M. Contreras, Alex López-Córdoba, Edwin Yesid Gómez-Pachón, Julián Carrera, María Eugenia Suárez-Ojeda

**Affiliations:** 1GENOCOV Research Group, Department of Chemical, Biological and Environmental Engineering, School of Engineering, Universitat Autònoma de Barcelona, Escola d’Enginyeria. Edifici Q Campus UAB, Bellaterra, 08193 Barcelona, Spain; 2División Catalizadores y Superficies, Instituto de Investigaciones en Ciencia y Tecnología de Materiales, INTEMA (CONICET), Av. Colón 10850, Mar del Plata 7600, Argentina; 3Grupo de Investigación en Bioeconomía y Sostenibilidad Agroalimentaria, Escuela de Administración de Empresas Agropecuarias, Facultad Seccional Duitama, Universidad Pedagógica y Tecnológica de Colombia, Carrera 18 con Calle 22, Duitama 150461, Colombia; 4Grupo de Investigación en Diseño, Innovación y Asistencia Técnica de Materiales Avanzados-DITMAV, Escuela de Diseño Industrial, Universidad Pedagógica y Tecnológica de Colombia-UPTC, Duitama 150461, Colombia

**Keywords:** polyhydroxyalkanoates, mixed microbial cultures, extraction, sodium hypochlorite, dimethyl carbonate, chloroform

## Abstract

The use of mixed microbial cultures (MMC) and organic wastes and wastewaters as feed sources is considered an appealing approach to reduce the current polyhydroxyalkanoates (PHAs) production costs. However, this method entails an additional hurdle to the PHAs downstream processing (recovery and purification). In the current work, the effect of a sodium hypochlorite (NaClO) pre-treatment coupled with dimethyl carbonate (DMC) or chloroform (CF) as extraction solvents on the PHAs recovery efficiency (RE) from MMC was evaluated. MMC were harvested from a sequencing batch reactor (SBR) fed with a synthetic prefermented olive mill wastewaster. Two different carbon-sources (acetic acid and acetic/propionic acids) were employed during the batch accumulation of polyhydroxybutyrate (PHB) and poly(3-hydroxybutyrate-co-3-hydroxyvalerate) (PHBV) from MMC. Obtained PHAs were characterized by ^1^H and ^13^C nuclear magnetic resonance, gel-permeation chromatography, differential scanning calorimetry, and thermal gravimetric analysis. The results showed that when a NaClO pre-treatment is not added, the use of DMC allows to obtain higher RE of both biopolymers (PHB and PHBV), in comparison with CF. In contrast, the use of CF as extraction solvent required a pre-treatment step to improve the PHB and PHBV recovery. In all cases, RE values were higher for PHBV than for PHB.

## 1. Introduction

There is a growing environmental concern as a result of the problems driven from the overuse of petroleum-based plastics. This has caused enormous interest in the research of new bio-based and biodegradable materials that can potentially replace them. According to Kurian et al. [1], as of 2020, the estimated market size for bioplastics and biopolymers in North America and Europe was around 10.5 billion USD, which is projected to rise to 25–30 billion USD by 2025.

Polyhydroxyalkanoates (PHAs) are one of the most promising bioplastics because they are biodegradable and biocompatible and can be obtained from renewable resources. PHAs possess similar mechanical, thermal, and barrier properties to petroleum-based plastics such as polyethylene and polypropylene [2,3]. PHAs have been used in several fields including agriculture, medicine, packaging, and pharmacy [1,4,5].

One of the most important limitations for the commercial application of PHAs is their high production costs compared with petroleum-based plastics [1]. The current price of PHAs is between 2.2 and 5.0 €/Kg. This is around three times higher than the price of petroleum-based plastics such as polystyrene (PS), polypropylene (PP), or polyvinyl chloride (PVC) (<1.0 €/kg) [6]. The main factors affecting the final cost of PHAs are the use of pure cultures (e.g., genetically modified bacterial strains), specific carbon sources and operating conditions, and downstream processing [7].

Currently, a large part of the industrial scale production of PHAs is achieved using pure or genetically modified cultures, which necessitates sterile conditions and the utilization of expensive substrates [8,9]. Therefore, ongoing efforts to lower the cost of PHA manufacturing relies on using MMC. MMC do not require sterile conditions and are flexible enough to adjust to variations in the carbon source, affording the use of a wide variety of waste substrates, such as agricultural and industrial wastes [10,11]. In particular, MMC from waste-activated sludge from wastewater treatment plants have demonstrated a great PHA-accumulating potential under stressing conditions [10,11]. However, it has been reported that different substrates and production methods lead to various types of microbial biomass. Thus, an extraction method that is optimized for one particular MMC may not be effective in all conditions [12]. In addition, the recovery of PHAs from MMC is a more complex process because several microorganisms are coexisting and the different cellular matrices that surround the PHAs’ granules hinders their extraction because the cells present different resistance to lysis [13,14]. Therefore, it is often necessary to apply pre-treatment methods that favor cell disruption, such as NaClO digestion [10,15].

Extraction using chlorinated solvents, such as chloroform (CF) and 1, 2-dichloroethane, is one of the most widespread methods to recover PHAs from MCC [7]. In contrast, the use of halogen-free solvents, such as dimethyl carbonate (DMC), in the extraction of PHAs from MMC, has been largely unexplored [7]. DMC has the advantage of being considered a green solvent because it is fully biodegradable, as well as being minimally toxic for the operators and the environment [7]. Samorì et al. (2015) [13] studied the extraction of PHAs from MMC using a NaClO pre-treatment coupled with DMC, finding that polymer recovery increased from 50% to 80%. Recently, Elhami et al. (2022) [16] extracted PHBV from MMC using DMC as an extraction solvent, and ethanol to precipitate the dissolved PHA. They found that the highest PHBV recovery value was 94%. This occurred when biopolymer precipitation was performed following complete evaporation of DMC.

In the current work, the application of an MMC pre-treatment using NaClO as an alternative to increase the extraction of PHAs with DMC or CF was investigated. The extracted biopolymers were characterized in terms of polymer structure, molecular weight, and thermal properties.

## 2. Materials and Methods

### 2.1. Production of PHAs

MMC used in the present study were harvested from a sequencing batch reactor (SBR) fed with a synthetic prefermented olive mill wastewaster (Montiel-Jarillo et al., submmited) [17]. The SBR was operated under fully aerobic conditions and under a feast/famine regimen. The biomass was collected at the end of the feast phase. The supernatant was removed by settling, and the biomass was resuspended in a mineral medium devoid of a nitrogen source, as described in Montiel-Jarillo et al. [18]. Before the accumulation experiments, the biomass was aerated overnight in the absence of any substrate. Then, batch PHA-accumulation experiments were performed in 1 L glass stirred reactors according to the protocol described in Mannina et al. (2019) [19]. Briefly, the change of the dissolved oxygen (DO) concentration in the reactor was used as an indicator of the substrate exhaustion. The experiment started with the addition of a pulse of the substrate (0.2 gCOD L^−1^). The addition of the substrate caused a noticeable drop in the DO. After a certain time (typically, 5–10 min), the DO increased due to the substrate depletion. Then, a new pulse of the substrate was added. Using this procedure, a total of 30 pulses of the substrate were added to the reactor in 4–5 h of operation. Two different carbon sources were used. In batch A, acetic acid (HAc) was used to obtain PHB, whereas in batch B a mixture of HAc and propionic acid (HPr) in a proportion of 75:25 over the chemical oxygen demand (COD) leads to the accumulation of the copolymer PHBV [18].

During PHA-accumulation experiments, mixed liquor samples were taken to determine the concentration of biomass as total and volatile suspended solids (TSS, VSS), acetate and propionate as volatile fatty acids (VFA), and the specific content of PHA of the biomass. TSS and VSS concentrations were measured according to the APHA methods 2540D and 2540E, respectively [18]. The samples for VFA were filtered through 0.45 μL pore size membranes and transferred to 2 mL septum-capped vials. VFA concentration of the filtrate was measured according to the procedure described by Montiel-Jarillo et al. [18]. To evaluate the specific content of PHA in the biomass, 10 mL of the culture was mixed with 0.4 mL of formaldehyde (37 wt% in H_2_O) to inhibit biological activity. Then, at the end of each experiment, 6 mL of formaldehyde (37 wt% in H_2_O) was added to the reactor to inhibit biological activity. In all cases, the biomass was harvested by centrifugation at 11× *g* for 45 min. The supernatant was discarded. The PHA-enriched biomass was stored overnight at −80° and then lyophilized during 24 h.

The amount of PHA in the biomass was quantified by gas chromatography (GC) [18]. A known amount of the lyophilized biomass was mixed with benzoic acid as an internal standard. Then, 1.5 mL of butanol and 0.5 mL of hydrochloric acid were added, and the mixture was incubated at 100 °C for 8 h. After that, 2.5 mL of hexane and 4 mL of Milli-Q were added. The tubes were vortexed and left to stand for 15 min to allow the separation of the phases. The organic phase was transferred to clean tubes and 4 mL aliquot of Milli-Q water was added. Then, the tubes were vortexed and centrifuged at 2× *g* for 10 min. The organic phase was filtered through 0.22 μm filters and transferred into GC vials. One microliter of the sample was injected and analyzed in an Agilent Technologies (7820 A) gas chromatograph equipped with an FID detector and an HP-InnoWax column (30 m × 0.53 mm × 1 μm).

### 2.2. Evaluation PHAs Extraction

PHAs were extracted from the lyophilized biomass using dimethyl carbonate (DMC) and chloroform (CF) as the extraction solvents, according to the method described by Samorì et al. [20]. The PHAs extraction assays were carried-out at boiling temperature of each solvent (DMC at 90 °C and CF at 60 °C) for 1 h, using a lyophilized biomass concentration of 50 g dry-cell/L. These operational conditions were chosen from preliminary experiments in which different biomass concentrations (25 and 100 g dry-cell/L) and extraction time (1 and 3 h) were evaluated (Appendix A).

Figure 1 shows the schematic representation of the proposed PHAs extraction process using DMC and CF. Fifty milligrams of the lyophilized biomass were transferred to Eppendorf vials. Then, the samples were resuspended in 1 mL of Milli-Q and mixed with 2 mL of the tested extraction solvent. The vials were periodically stirred using a vortex. Then, vials were centrifuged at 6.5× *g* for 15 min at room temperature. Three phases were obtained: (1) a supernatant phase composed by water and water-soluble compounds, (2) an intermediate phase that corresponds to the organic fraction containing the solubilized PHAs, and (3) a precipitate composed by biomass and cell debris. The organic fraction (2) was recovered and filtered through 0.45 µm pore size cellulosic membranes (Millipore^®^, Burlington, MA, USA) to remove any cell debris. Finally, the solvent was removed by evaporation to obtain a dried PHA sample.

A second set of experiments to enhance the recovery of PHA from the lyophilized biomass were performed employing a chemical digestion using sodium hypochlorite (NaClO) [13]. In this case, 50 mg of lyophilized biomass was resuspended in 5 mL of NaClO 5% (*w*/*v*) and incubated at 100 °C for 1 h. The digested samples were centrifuged at 3× *g* for 10 min at room temperature. The pellets were washed three times with Milli-Q water and resuspended in 1 mL of Milli-Q water. Then, PHAs were extracted using DMC and CF according to the procedure previously described in this section.

#### Calculations

To compare the different tested extraction protocols, the overall recovery efficiency (RE, %) was calculated as follows:(1)$$\mathrm{RE}=100\frac{Y_{E}P_{E}}{PHA_{content}}$$
where *Y_E_* (g extract/g dry-cell) is the extraction yield, *PHA_content_* (g PHA/g dry-cell) is the amount of PHA (measured by GC) within the cells, and *P_E_* (g PHA/g extract) is the amount of PHA (measured by GC) per gram of extract. Measured purities were 0.79 g PHA g extract^−1^ for extracts from batch A, and 0.85 g PHA g extract^−1^ for the copolymer obtained from samples of batch B.

### 2.3. Characterization of the Obtained Biopolymers

To obtain enough amounts of PHAs for their further characterization, PHAs were extracted from the lyophilized biomass using a Soxhlet device (Buchi, E-816 SOX). In these cases, 2 g of the lyophilized biomass obtained at the end of each PHA-accumulation experiment was mixed with 220 mL of NaClO 5% (*w*/*v*) at 100 °C for 1 h. The digested sample was centrifuged at 3× *g* for 20 min. The obtained pellet was centrifuged and washed three times with milli-Q water and introduced into the Soxhlet cartridge. PHA was extracted using 90 mL of chloroform (CF) for 1 h. Finally, CF was removed by evaporation.

The chemical structure of the obtained PHAs was determined by quantitative ^1^H and ^13^C nuclear magnetic resonance (NMR) spectra using a BRUKER DRX-500 spectrometer. Known amounts (30–40 mg) of the obtained PHA were dissolved in deuterated chloroform (CDCl_3_). The software Bruker TopSpin3.5pl7 was used to analyze the obtained NMR spectra. Also, gradient-selected ^1^H/^13^C heteronuclear single quantum coherence (HSQC) spectra were acquired.

Mass-average (M_w_) and number-average (M_n_) molecular weights of the biopolymers were measured by Gel Permeation Chromatography (GPC) on a Waters equipment provided with RI and UV detectors. Biopolymer samples were diluted (0.1% *w*/*v*) and filtered. Then, 100 µL of this solution were injected and operated using 1,1,1,3,3.3-hexafluoro-2-propanol as a mobile phase with a flow of 0.5 mL min^−1^. HHR5E and HR2 Waters linear Styragel columns (7.8 mm × 300 mm, pore size 103–104 Å) packed with crosslinked polystyrene and protected with a pre-column were used. Molecular weight was calibrated using poly(methyl methacrylate) as the reference compound.

To evaluate the degradation temperature (T_d_), melting temperature (T_m_), and enthalpy of fusion (Δ*H_m_*) of the obtained biopolymers, a simultaneous Thermogravimetry (TGA)–Differential Scanning Calorimetry (DSC) analysis (Netzsch STA 449F1 Jupiter^®^, Gaithersburg, MD, USA) under a nitrogen atmosphere was performed [21]. The crystallinity degree (X, %) of the biopolymer was calculated as follows Dai et al. [22].
(2)$$X\;\left(\%\right)=100\frac{\Delta H}{\Delta H_{PHB}^{o}}$$
where Δ*H* (J/g) is the melting enthalpy of the analyzed PHA, and $\Delta H_{PHB}^{o}$ = 146 J/g is the melting enthalpy of a fully crystalline PHB [23].

## 3. Results and Discussion

### 3.1. Production of PHA Using Mixed Microbial Cultures (MMC)

Figure 2 shows a typical result corresponding to batch A. When a pulse of HAc was added, pH immediately dropped from a value above 8 to about 6. Also, a quite noticeable decrease of the DO concentration was obtained. While a gradual increase in pH was observed, DO remained below 1 mg L^−1^ for about 6–8 min. Then, a sudden increase in DO indicated the substrate depletion. At this point, the measured soluble COD was negligible, confirming the absence of a substrate. Therefore, a new pulse of HAc was added. Figure 2 also shows that the specific content of PHA in the cells increased as a function of the added pulse.

When HAc was tested as the sole carbon source, the specific PHA production rate (*q_PHA_*), and the specific substrate consumption rate (*q_S_*) were 0.37 C-mol PHA (C-mol biomass)^−1^ h^−1^ and 0.52 C-mol VFA (C-mol biomass)^−1^ h^−1^, respectively. Accordingly, the PHA yield (Y_PHA_) was 0.71 C-mol PHA (C-mol VFA)^−1^. In other words, about 70% of the carbon in form of HAc yielded PHA, while the other 30% was used for biomass synthesis and energy production (e.g., respiration). At the end of the experiment, the specific PHA content of the biomass was about 0.57 gPHA g(dry-cell)^−1^. CG analysis showed that more than 95% (*w*/*w*) of the PHA corresponded to HB units, suggesting that the obtained polymer was mainly PHB.

Figure 3 shows a PHA accumulation assay using a mixture of HAc:HPr as the carbon source (batch B). A similar trend with respect to pH, and DO in comparison with batch A was obtained (Figure 2), however, the PHV increased by up to 35% *w*/*w*. In this case, *q_PHA_* and *q_S_* values were 0.47 C-mol PHA (C-mol biomass)^−1^ h^−1^ and 0.55 C-mol VFA (C-mol biomass)^−1^ h^−1^, respectively. According to these values, Y_PHA_ corresponding to these experiments was 0.85 C-mol PHA (C-mol VFA)^−1^. Thus, the performance of the mixture of HAc:HP, with reference to the production of PHA, was slightly better than that of HAc as the sole carbon source. CG analysis showed that the polymer composition was quite constant during the whole experiment. The average polymer composition of the obtained PHA was 52% (*w*/*w*) of HB and 48% (*w*/*w*) of HV. At the end of the experiment, the PHA content of the biomass reached 0.72 gPHA g dry-cell^−1^.

Kinetic (*q_PHA_*, *q_S_*) and stoichiometric (*Y_PHA_*) coefficients obtained for both tested carbon sources were within the range reported for the production of PHA using MMC grown on different synthetic substrates such as acetate, propionic, or mixtures of them [24,25,26]; fermented sugar cane molasses; cheese whey; and waste-activated sludge [27,28].

### 3.2. Evaluation of PHAs Extraction

Table 1 shows the obtained PHA recovery efficiencies (RE) with DMC or CF as extraction solvent using a biomass concentration of 50 g/L and extraction time of 1 h. In all cases, it was observed that PHBV had higher recovery than PHB, suggesting that PHBV was longer soluble in both solvents. In addition, it was found that when a MMC pre-treatment was not carried out, the use of DMC solvent afforded higher PHB and PHBV recovery than CF. This behaviour was probably because the use of a solvent with a high boiling point promoted cell lysis, leading to better contact of the solvent with the intracellular granules of PHAs. On the other hand, it was observed that the pre-digestion step successfully enhanced the PHB and PHBV recovery using both solvents. It has been stated that NaClO weakens cell membranes, thus facilitating the subsequent extraction of PHA by solvents [7,8].

In complementary experiments, the effect of different biomass amounts (25, 50, and 100 g dry-cell/L) and incubation times (1 and 3 h) on PHB and PHBV recovery from MMC was evaluated for comparison purposes. Furthermore, the influence of the ethanol-induced precipitation on the PHAs extraction was studied (Appendix A). It was found that when the lyophilized biomass concentration was 25 g/L, the PHB and PHBV recoveries were negligible. The use of concentrations of 100 g/L led to lower recovery efficiencies than those obtained when using a concentration of 50 g/L. This may be due to the suspension at 100 g/L becoming very viscous, leading to quite low recovery of the biopolymers [29]. Besides, the results showed that an increase in the extraction time from 1 to 3 h and a precipitation step using ethanol did not improve the recovery of the polymers in comparison with the results showed in Table 1. Similarly, de Souza Reis et al. (2020) [29] studied the effect of different extraction times (0.25, 0.5, 1, 1.5, and 2 h) and biomass to DMC solvent proportions (1, 2.5, 5, and 10%) on the PHBV recovery from MMC. They obtained recovery percentages of 32.9 when using a biomass to solvent ratio of 1% with an extraction time of 1.5 h. Abbasi et al. (2022) [30] extracted PHBV from MMC using CF and DMC as extraction solvents, obtaining recovery values of 33.5% and 30.6%, respectively. This variability observed in the recovery percentage of PHBV could be due to the heterogeneity and complex cell structure of the MMCs, meaning that a specific extraction method may not have been efficient for all MMC sources [26,29].

### 3.3. Characterization of the Obtained Biopolymers

#### 3.3.1. Quantitative ^1^H/^13^C Nuclear Magnetic Resonance (NMR)

Figure 4a shows the ^1^H-spectrum corresponding to the biopolymer obtained from batch A (PHB), exhibiting three characteristic signals of PHB [31]. The signal corresponding to the methyl group (peak 4 in Figure 4a) is the first doublet at 1.28 ppm (B4 in Figure 4a). Signals at 2.50–2.58 ppm (peak 2 and peak 3 in Figure 4a) correspond to the methylene group (B2 in Figure 4a), while the multiplet at 5.25 ppm (peak 1 in Figure 4a) corresponds to the methine group (B3 in Figure 4a). Other small signals observed in Figure 4a may be due to the presence of small amounts of HV. Figure 4b shows the ^1^H-spectrum corresponding to the biopolymer from batch B (PHBV). According to the literature, this spectrum has the characteristic signals of a copolymer PHBV ^44^. Methyl groups corresponding to HV (V5) and HB (B4) are represented by the resonance peaks at 0.90 (peak 8 in Figure 4b) and 1.28 ppm (peak 7 in Figure 4b), respectively. The peak at 1.63 ppm (peak 6 in Figure 4b) is due to the HV methylene protons (V4). Overlapping resonance peaks within the range of 2.50–2.60 ppm (peaks 3, 4, 5 in Figure 4b) represent the methylene protons of HV (V2) and HB methylene group (B2). Resonance signals at 5.15 and 5.25 ppm (peaks 1,2 in Figure 4b) are characteristic for the methine proton of HV (V3) and HB (B3), respectively.

The molar fraction of HB and HV corresponding to the polymer obtained from batch B can be estimated by the intensity of the signals B3 with respect to V3, and V5 to B4 [32] (Table 2). According to the data corresponding to the methine groups, the obtained copolymer was composed of 58% of HB and 42% of HV on a molar basis. A similar composition was obtained from the intensity of the methylene groups (59% and 41% of HB and HV, respectively). It must be noted that these results are similar to those obtained by GC during the accumulation assay (Figure 3), confirming that the biopolymer obtained from batch B was a copolymer of PHBV.

Figure 5 shows resonance ^13^C-spectra corresponding to the obtained polymers. The results for the resonance peaks were similar to those documented in the literature [14,32,33,34]. Only four clearly discernible peaks can be observed in the ^13^C-spectrum corresponding to PHB (Figure 5a). Those peaks correspond to carbonyl (B1) at 169.12 ppm, methyl carbon (B4) at 19.77, and methylene (B2) and methine (B2) at 40.78 and 67.62 ppm, respectively. All those peaks are characteristic of PHB. Conversely, the ^13^C-spectrum corresponding to PHBV (Figure 5b) was typical of a copolymer PHBV. Figure 6 shows an expansion of several zones corresponding to the ^13^C-spectrum depicted in Figure 5b. The methylene moieties of HB and HV are responsible for the peaks between 26 and 41 ppm.

Carbonyl peaks can be observed at around 169 ppm [14,22,34]. These signals contribute to diads of HB and HV units, namely BB, BV, VB, or VV, where B corresponds to butyrate, and V to valerate. The peak at 169.15 ppm is consistent with the carbonyl resonance of PHB, and it was assigned to the sequence BB. According to Doi et al. [35], the sequence VV was assigned for the peak that appears with a difference of 0.38 ppm. Intermediate peaks with low intensities were assigned to the BV and VB sequences. The obtained peaks within the methylene region were consistent with those reported in the literature. First, the peak at 40.78 ppm that corresponds to the methylene of the HB unit (B2) exhibits a shoulder due to the presence of two almost overlapping peaks. The split is due to the diad sequences BV and BB. For the HV unit, the signals for the side-chain methylene group (V4) and main-chain methylene group (V2) may be differentiated, unless both signals are split into four peaks and have almost similar intensities. The main-chain methylene signal is observed between 26.75 and 26.85 ppm and the shift for side-chain methylene is around 38.64–38.79 ppm. In both cases, peaks were assigned for triad sequences of VVV, BVV, VVB, and BVB.

The molar ratio of HV with respect to HB in the copolymer can be obtained from the signals corresponding to the methine carbons at 67.61 ppm for PHB (B3 in Figure 5a), and at 71.91 ppm for PHV (V3 in Figure 5b) [14]. According to these considerations, PHBV was composed of 58.4% of HB and 41.6% of HV. The calculated composition was in agreement with that obtained from the ^1^H spectrum. Results obtained from the HSQC spectra confirmed the assignment of resonance peaks of the carbons belonging with their corresponding hydrogen signals. The C-H couplings in HSQC are shown in Figure 7 for PHB, and Figure 8 for PHBV.

The comonomer composition distribution (CCD) of a copolymer has been identified as a key factor in determining the physical properties of a biopolymer [34]. To evaluate the degree of randomness of the obtained polymer in batch B, ^1^H NMR (Figure 4b) and ^13^C NMR (Figure 5b), spectra were analyzed using the procedure proposed by Kamiya et al. [36] and other authors [34,37]. Two parameters, D and R, can be calculated from the relative peak intensities of ^1^H and ^13^C NMR spectra. Statistically random copolymers have D values in the range 0.99–1.5, while D > 1 or D < 1 define non-random copolymers. Besides, a D value close to 0 is characteristic for alternating nature copolymers, while D > 1 indicates “blocky” copolymers [21,36]. However, in cases where D is greater than 1, the copolymer may be a true block copolymer, a mixture of random copolymers, or a mixture of HB and HV homopolymers [36,37]. It must be noted that according to several authors, the parameter R is more sensitive than D for estimating the degree of randomness of a copolymer [34,37]. A value of *R* around 0 indicates a diblock copolymer, while an R value of 1 corresponds to a completely random distribution [21,34,37].

Table 3 shows the relative peak intensities of ^13^C NMR spectra corresponding to the copolymer PHBV. These were used to calculate the parameter D and R. For the PHBV obtained in this study, the calculated value of D was 4.06. Thus, it can be concluded that the microstructure corresponds to a “blocky” copolymer character. Considering reports found in the literature, PHBV may be considered as a mixture of random copolymers, or a simple block structure [21,34]. Moreover, R value obtained herein was 0.65, supporting the idea that the obtained PHBV is closer to a “blocky” copolymer rather than a random copolymer. As a general rule, “blocky” materials have better elongation properties (e.g., larger Young’s module and tensile strength) than random polymers [38,39].

#### 3.3.2. Gel-Permeation Chromatography (GPC)

The results obtained regarding the mass-average molecular weight (M_w_), number-average molecular weight (M_n_), and polydispersity index (PDI) are summarized in Table 4. These parameters are of great importance as they are responsible for the end-use suitability of a given polymer for specific applications [37,40].

The corresponding M_w_ and M_n_ of the polymers obtained in this study were lower than PHA molecular weights frequently reported [40]. The reason for such low molecular weights may be a consequence of the effect of different conditions during the extraction process. On one hand, the first extraction step was the biomass digestion using NaClO for 1 h at 100 °C. Although these conditions are favorable for disrupting cell walls, they also may favor polymer hydrolysis, yielding a decrease of the molecular weight of the obtained polymer [13,41]. In this sense, several authors report that amorphous polymers are weak to alkaline saponification [42]. However, other authors support the use of NaClO in the presence of an extraction solvent, arguing that once PHA is released, it may be immediately dissolved in the extraction solvent, preventing the polymer hydrolysis [42]. Moreover, the temperature and time of the thermal treatment also affect the molecular weight of the obtained polymer. As a general rule, higher temperatures increase the PHA solubility, but also favor its hydrolysis [40]. Additionally, other variables, such as the extraction from dried or wet cells, and fermentation conditions (pH, type and concentration of the carbon source, and nutrients) were reported as factors that affect the molecular weight of the obtained biopolymers [14,37,43].

The polydispersity index (PDI) of a given polymer is a measure of the heterogeneity of the polymer chain lengths [40]. In principle, PDI values range from one to infinity. A polymer composed of molecules with the same chain lengths has PDI values close to one. Conversely, PDI > 1 are characteristic of polymers composed of different chain lengths. While for a typical addition polymerization, PDI can range around 5–20, most probable PDI values for typical step polymerizations are around two. As a general rule, polymers with PDI close to one are preferred, which enables their use in a huge range of applications [41]. Although molecular weights of the obtained polymers were lower than those commonly found in the literature, their PDI values (Table 4) were within the range reported (from 1.84 to 7.12) by other authors during PHAs extraction from MMC [10,21].

#### 3.3.3. Thermogravimetry (TGA) and Differential Scanning Calorimetry (DSC)

Figure 9 shows that TGA profiles corresponding to both obtained polymers were similar. The degradation of both polymers occurred as a single step, starting at around 200 °C. In both cases, the temperature corresponding to the maximum degradation rate was 291 ± 1 °C. The degradation temperature for a 5% weight loss obtained for both polymers was 243 ± 2 °C. This value was within the range reported by other authors [21,23,42].

Table 5 shows the results obtained with the DSC corresponding to the obtained polymers. PHB exhibited a single melting point at 154.6 °C. This value is within the range of results reported in several studies [42,43,44]. Conversely, for PHBV, two melting points were observed at 78.3 and 152.9 °C. These melting points were close to those reported by Arcos-Hernández et al. [21] (77.5 and 159.2 °C) corresponding to a PHBV obtained using the same feeding strategy and carbon source as in the experiments reported herein. According to several authors [21,36], blocky copolymers with D > 1.5, such as the PHBV obtained in the present study, can exhibit two or three melting points as a result of the presence of polymers with different structures.

Based on the melting enthalpy for each biopolymer, the degree of crystallinity (*X*_c_) was obtained using Equation (2). Table 5 shows that the degree of crystallinity corresponding to PHBV was 2.4 times lower than that corresponding to PHB. According to several authors, polymers with higher crystallinity have a greater range of potential applications [23,45].

## 4. Conclusions

In this study, PHB and PHBV were obtained from MMC using acetic and a mixture of acetic/propionic acids as carbon sources. Overall, the recovery of PHBV was higher than PHB, regardless of which extraction solvent was used. The PHAs recovery without a NaClO pre-treatment step was highest when DMC was employed, while when CF was used, a pre-treatment step was necessary to improve the extraction of PHB and PHBV. PHAs characterisation by ^1^H and ^13^C NMR spectra demonstrated that PHBV corresponded to a “blocky” copolymer. In general, thermal properties suggested that the presence of HV units confers desirable thermal characteristics for further PHA processability. The decomposition temperature of both polymers was similar, but the melting point and degree of crystallinity were lower in PHBV than that of PHB. Further studies are necessary to microbiologically characterize the MMC obtained.

## Figures and Tables

**Figure 1 polymers-14-03938-f001:**
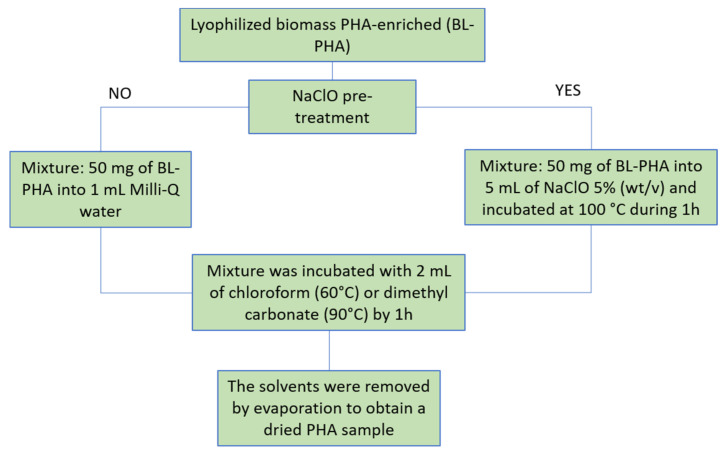
Schematic representation of the proposed PHAs extraction process using DMC and CF.

**Figure 2 polymers-14-03938-f002:**
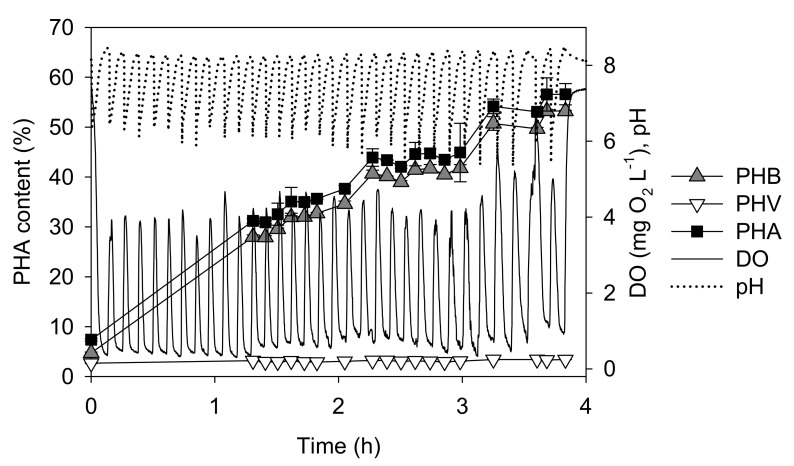
Performance of batch A as a function of time using HAc as the sole carbon source. The black line represents the DO profile, while the black dotted line corresponds to pH. The lines connecting the experimental points were included as a visual aid. Bars indicate the standard deviation of triplicates of PHAs content.

**Figure 3 polymers-14-03938-f003:**
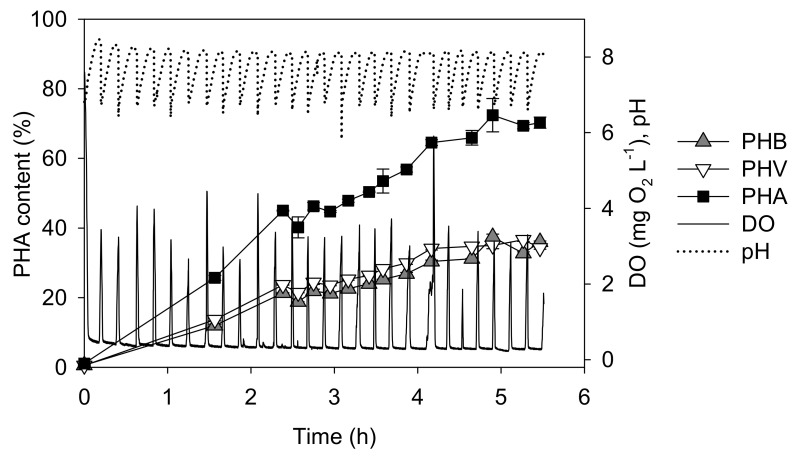
Performance of batch B as a function of time using a mixture of HAc and HPr. The black line represents the DO profile, while the black dotted line corresponds to pH. The lines connecting the experimental points were included as a visual aid. Bars indicate the standard deviation of triplicates of PHAs content.

**Figure 4 polymers-14-03938-f004:**
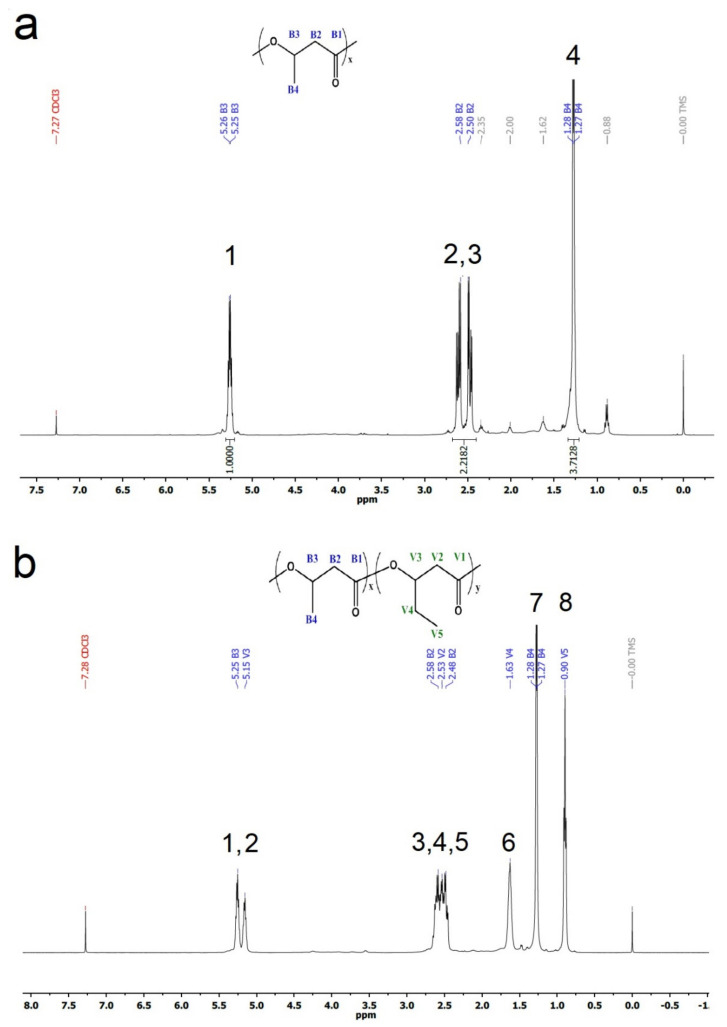
^1^H-NMR spectra corresponding to PHB (**a**) and PHBV (**b**), obtained from batch A and B, respectively.

**Figure 5 polymers-14-03938-f005:**
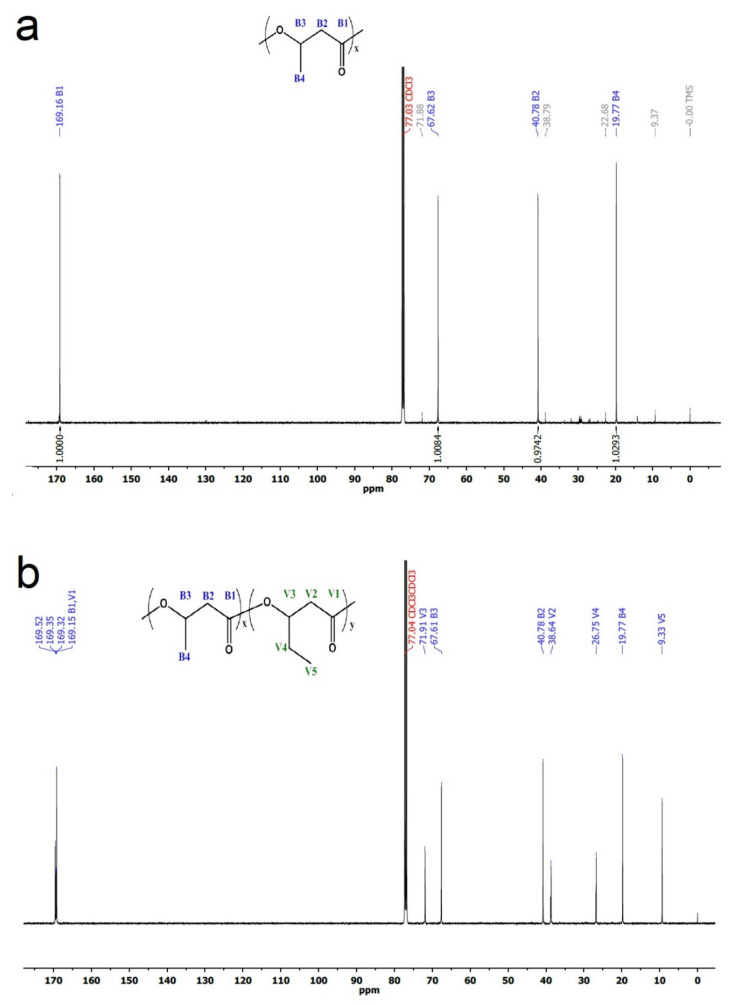
^13^C-NMR spectra corresponding to PHB (**a**) and PHBV (**b**) obtained from batch A and B, respectively.

**Figure 6 polymers-14-03938-f006:**
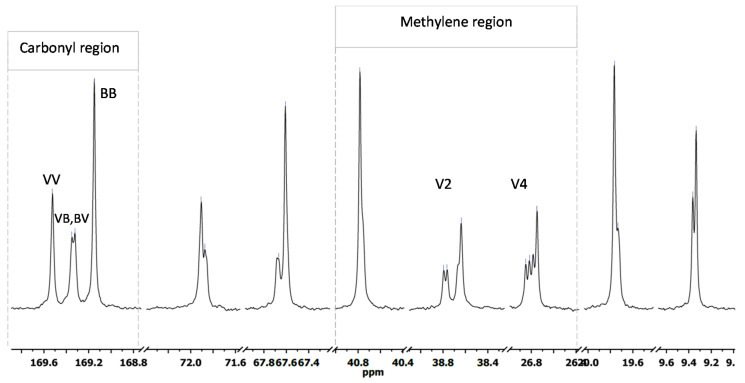
Detail of ^13^C-NMR spectrum corresponding to PHBV obtained from batch B.

**Figure 7 polymers-14-03938-f007:**
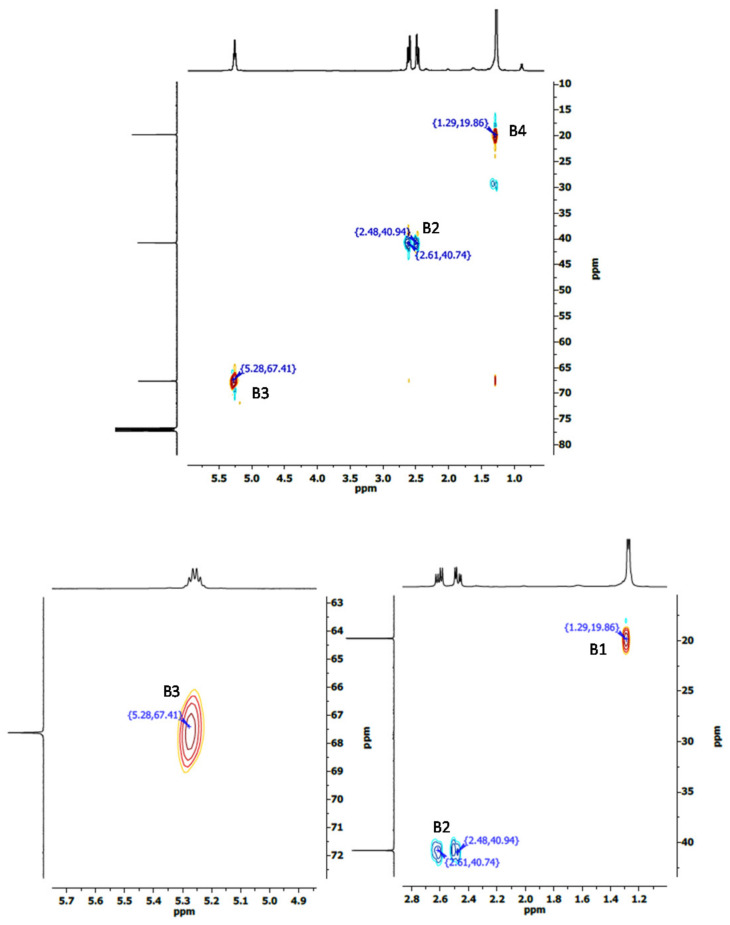
HSQC spectra corresponding to PHB obtained from batch A.

**Figure 8 polymers-14-03938-f008:**
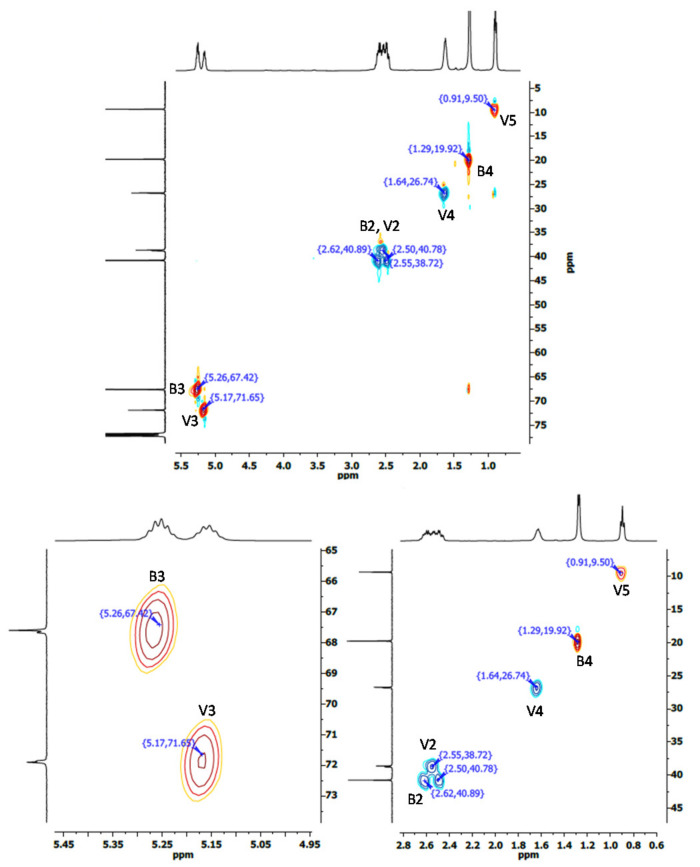
HSQC spectra corresponding to PHBV obtained from batch B.

**Figure 9 polymers-14-03938-f009:**
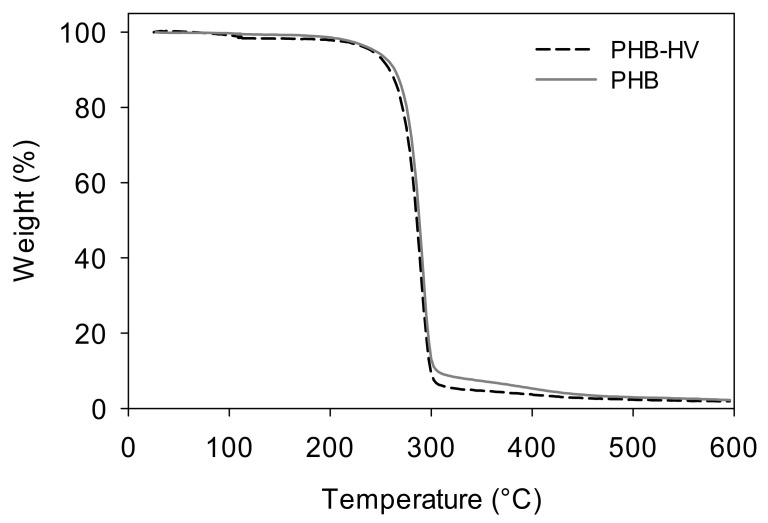
Thermogravimetric curves corresponding to PHB (continuous line) and PHBV (dotted line) obtained from batch A and B, respectively.

**Table 1 polymers-14-03938-t001:** PHA recovery efficiencies (RE) using dimethyl carbonate (DMC) or chloroform (CF).

Sample	Pre-Treatment with NaClO	RE (%)	
		DMC (at 90 °C)	CF (at 60 °C)
Biomass enriched in PHB	No	4 ± 3	2 ± 1
	Yes	8 ± 3	17 ± 1
Biomass enriched in PHBV	No	17 ± 1	5 ± 2
	Yes	25 ± 4	23 ± 2

Depicted values correspond to the average ± one standard deviation of three measurements.

**Table 2 polymers-14-03938-t002:** Assignment of resonance peaks for ^13^C-NMR spectra of copolymer PHBV.

Position	Diad/Triad	^13^C (ppm)
**V5**	VV	9.33
	VB	9.36
**B4**	BB	19.73
	BV	19.77
**V4**	VVV	26.75
	BVV	26.77
	VVB	26.82
	BVB	26.85
**V2**	VVV	38.64
	BVV	38.65
	VVB	38.76
	BVB	38.79
**B2**	BV	40.77
	BB	40.78
**B3**	VB	67.61
	VV	67.67
**V3**	BB	71.87
	BV	71.91
**B1**	BB	169.15
	BV	169.32
**V1**	VB	169.35
	VV	169.52

B and V represent butyrate and valerate units, respectively.

**Table 3 polymers-14-03938-t003:** Experimental monomer, dyad, and triad sequence mole fractions obtained from ^1^H^a^ and ^13^C^b^ NMR spectra corresponding to the copolymer PHBV.

*Monomer*	F_B_	*0.587 ^a^* *0.584 ^b^*
	F_V_	*0.413 ^a^* *0.416 ^b^*
*Dyad*	F_BB_	*0.409*
	F_BV_	*0.176*
	F_VB_	*0.151*
	F_VV_	*0.264*
*Triad*	F_VVV_	*0.181*
	F_VVB_	*0.081*
	F_BVV_	*0.075*
	F_BVB_	*0.075*

B: butyrate unit; V: valerate unit. ^*a*^ determined by ^1^H NMR spectra and ^*b*^ determined by ^13^C NMR spectra.

**Table 4 polymers-14-03938-t004:** Molecular weights corresponding to the obtained biopolymers.

PHA	M_w_ (×10^4^ Da)	M_n_ (×10^4^ Da)	PDI
PHB	4.53	1.69	2.7
PHBV	3.49	1.23	2.8

M_w_: mass-average molecular weight. M_n_: number-average molecular weight. PDI: polydispersity index.

**Table 5 polymers-14-03938-t005:** Thermal characterization parameters corresponding to the obtained biopolymers.

PHA	T_d-5%_	T_d-max_	T_m1_	ΔH_m1_	T_m2_	ΔH_m2_	X_c_ (%)
PHB	244.9		--	--	154.6	47.2	35.8
PHBV	242.3		78.3	15.9	152.9	3.44	14.7

T_5%_: degradation temperature corresponding to 5% weight loss. T_max_: temperature corresponding to the maximum degradation rate. X_C_: crystallinity (Equation (1)). Temperature values are expressed in °C and ΔH in J g^−1^.

## Data Availability

The data presented in this study are available on request from the corresponding author.

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
