# Peer review of "Improvement of the Polyhydroxyalkanoates Recovery from Mixed Microbial Cultures Using Sodium Hypochlorite Pre-Treatment Coupled with Solvent Extraction"

_polymers, 2022, doi:10.3390/polym14193938_

Round 1

Reviewer 1 Report

I think that the research article in present form is not adequate to be published in the journal; however, it is necessary to adjust throughout the text and restructured.

1.    The text present does not reflect the title, the title indicated an extraction process without contamination, however, the paper discusses production with 2 carbon sources and extraction with two solvents. Furthermore, the reported extraction efficiency is similar. It is necessary to align the content with the text, the title should reflect the findings obtained in the research.

2.    The introduction should also be contextualized to the content. In this case, if the term green solvent is maintained, the introduction should talk about this type of solvents, not only superficially.

3.    The information presented in the methodology is different from that presented in the summary. The methodology begins by stating that the inoculum was produced from synthetic olive mill waste what was the reason. The information is not congruent, from the title and abstract. It is not clear where the different carbon sources were used.

4.    Result section: A set of batch experiments under a feast/famine regime were performed in triplicates 224 using a PHA-enriched MMC as the inoculum. To produce PHB (batch A), acetic acid (HAc) 225 was used as the sole carbon source, this should be made clear in materials and methods.

5.    It is necessary that figure two is presented in a clear way, although it has a label, what is presented in the graph is incomprehensible, at least in the way it is presented.

6.    Figures 4 and 5 does not show the numbers of the peaks. Improve.

7.    The article contains a lot of information but lacks structure. The document should be restructured and written according to the objective, aligning all the sections since it is not clear what the contribution or novelty is.

8.    Approximately 40% of the references presented are more than 6 years old, it will be important to update the references, otherwise it would seem that it is a topic that has no future, if the document is updated it will reflect the importance of this. It should also include references from 2022, currently there are many publications that have been published recently.

9.    You must clarify what the contribution of the work is.

Reviewer 2 Report

Good and interesting work, demonstrating the possibility of eco-friendly DMC extraction of PHBV, as well as the copolymer synthesis using an acetate-propionate substrate; very thorough, detailed, and concordant analytical characterization (e.g., GC-NMR).

Recommendations: to go on the study with higher volume fermentation and biomass; I consider that the MMC should be somewhat microbiologically characterized (at least as bacterial genera present).

Little editing corrections:

line 51   North America and Europe instead of "North American and europe"

      57    have been  instead of "had been"

Chapter 2.1 of Materials and Methods should unitary use past tense

In 2.3 chapter title (line 178): delete "the"

In  table 3, PHBV instead of"PHBV)" (delete the bracket)

Reviewer 3 Report

The article gives a nice characterization of the PHB/PHBV that is produced in the batch experiments. However this is not new. Other also reported these types of characterization. However I do think this characterization is quite extensive. I do suggest that you try to condense the text and make it more to the point.

My main comment to the artivel is that the content of the article does not match with the title of the article. The use of Dimethyl carbonate as a green solvent is only a minor part of the article. Moreover the results of these experiments do not support the conclusions because of the low recovery rates. I think a better look needs to be taken at the reasons for this low recovery since other find much higher values.

Round 2

Reviewer 1 Report

Is it correct to use standard deviation to put error bars?

Author Response

Thank you for this advice. Error bars may show confidence intervals, standard errors, standard deviation or other quantifies. In this case, we chose to use the standard deviation because it is often used. This topic was clarified in the Figure legends.
On the other hand, according to the overall recommendations of the reviewer, we have improved the conclusions and the editing of the English language of the manuscript.

Reviewer 3 Report

The manuscript has improved since previous version. I still think that the results section can be been more concise.